# Fore-Mamba3D: Mamba-based Foreground-Enhanced Encoding for 3D Object Detection

**Zhiwei Ning[1,2], Xuanang Gao[1,2], Jiaxi Cao[1,2], Runze Yang[1,2,4],**
**Huiying Xu[5], Xinzhong Zhu[5,6], Jie Yang[1,2,3,*] & Wei Liu[1,2,3,*]**

[1] School of Automation and Intelligent Sensing, Shanghai Jiao Tong University
[2] Institute of Image Processing and Pattern Recognition, Shanghai Jiao Tong University
[3] Institute of Medical Robotics, Shanghai Jiao Tong University
[4] School of Computing, Macquarie University
[5] School of Computer Science and Technology, Zhejiang Normal University
[6] Beijing Geekplus Technology Co., Ltd
{zwning, fangkuar, caojiaxi, runze.y}@sjtu.edu.cn
{xhy, zxz}@zjnu.edu.cn, {jieyang, weiliucv}@sjtu.edu.cn
https://github.com/pami-zwning/ForeMamba3D

## Abstract

Linear modeling methods like Mamba have been merged as the effective backbone for the 3D object detection task. However, previous Mamba-based methods utilize the bidirectional encoding for the whole non-empty voxel sequence, which contains abundant useless background information in the scenes. Though directly encoding foreground voxels appears to be a plausible solution, it tends to degrade detection performance. We attribute this to the response attenuation and restricted context representation in the linear modeling for fore-only sequences. To address this problem, we propose a novel backbone, termed Fore-Mamba3D, to focus on the foreground enhancement by modifying Mamba-based encoder. The foreground voxels are first sampled according to the predicted scores. Considering the response attenuation existing in the interaction of foreground voxels across different instances, we design a regional-to-global slide window (RGSW) to propagate the information from regional split to the entire sequence. Furthermore, a semantic-assisted and state spatial fusion module (SASFMamba) is proposed to enrich contextual representation by enhancing semantic and geometric awareness within the Mamba model. Our method emphasizes foreground-only encoding and alleviates the distance-based and causal dependencies in the linear autoregression model. The superior performance across various benchmarks demonstrates the effectiveness of Fore-Mamba3D in the 3D object detection task.

## 1 Introduction

3D object detection is a critical task in computer vision with broad applications in autonomous driving (Mao et al., 2023; Grigorescu et al., 2020), embodied intelligence (Gupta et al., 2021; Huang et al., 2022; Zhang et al., 2025), and robotic navigation (Gul et al., 2019; Xu et al., 2023). Previous LiDAR-based methods (Shi et al., 2020b; Qi et al., 2017b; Li et al., 2021; Yan et al., 2018) achieve remarkable performance, which utilize sparse convolutional neural network (SpCNN) (Liu et al., 2015) or the Transformer (Vaswani, 2017) architecture as their backbones. However, the hardware incompatibility of SpCNN and the quadratic computational complexity of Transformer present substantial obstacles to their deployment in real-time detection applications. Recently, several Mamba-based methods (Gu & Dao, 2023; Liu et al., 2024a; Zhang et al., 2024c) empirically show that integrating a bidirectional scanning mechanism with the state space model (SSM) could achieve promising performance in 2D image recognition tasks with linear computational cost.

---

[*]The corresponding authors.

Inspired by the effectiveness in 2D computer vision tasks, some studies extend SSM to 3D object detection (Liu et al., 2024b; Zhang et al., 2024c). These methods can be briefly classified into group-based and group-free approaches. As shown on the left of Figure 1, group-based works (Liu et al., 2024b; Wang et al., 2023) partition the 3D voxel features into multiple groups along the X/Y-axis order for linear modeling. In contrast, group-free methods (Zhang et al., 2024c) directly flatten all the non-empty voxels in the scene via space-filling curves, such as Hilbert (Hilbert & Hilbert, 1935) or Z-order (Orenstein, 1986) curves. Though these methods encode the entire scene in various ways, the informative foreground occupies only a small portion, as illustrated on the right of Figure 1. This naturally motivates focusing on the effective foreground-only encoding technique.

Nevertheless, foreground-centric detectors confront inherent challenges. Primarily, imprecise or incomplete foreground sampling risks omitting critical structural information, thereby necessitating a specialized prediction and sampling strategy to ensure representational integrity. More critically, the sparse distribution of foreground voxels across distinct instances hinders conventional linear encoders from capturing long-range dependencies, resulting in response attenuation. Given that group-based methods excel at local modeling while group-free methods provide stronger global interaction, it is intuitive to mitigate response attenuation through a regional-to-global encoding strategy that leverages the advantages of both paradigms. In addition, enhancing the semantic and geometric awareness of the state variables in Mamba can further enrich contextual understanding and lead to superior overall performance.

In this paper, we propose Fore-Mamba3D to achieve foreground-enhanced encoding for 3D object detection. Specifically, for higher sampling accuracy, we predict the foreground score for each non-empty voxel and flatten the top-$k$ voxels into a 1D sequence via the Hilbert (Hilbert & Hilbert, 1935) space-filling curve template. However, directly applying the vanilla Mamba to foreground features leads to suboptimal performance due to response attenuation and insufficient textural representation. Therefore, we integrate the linear encoder with a regional-to-global sliding window (RGSW), which aggregates information from the regional patch to the entire sequence for sufficient long-range interaction. Then, we introduce the SASF-Mamba component for better

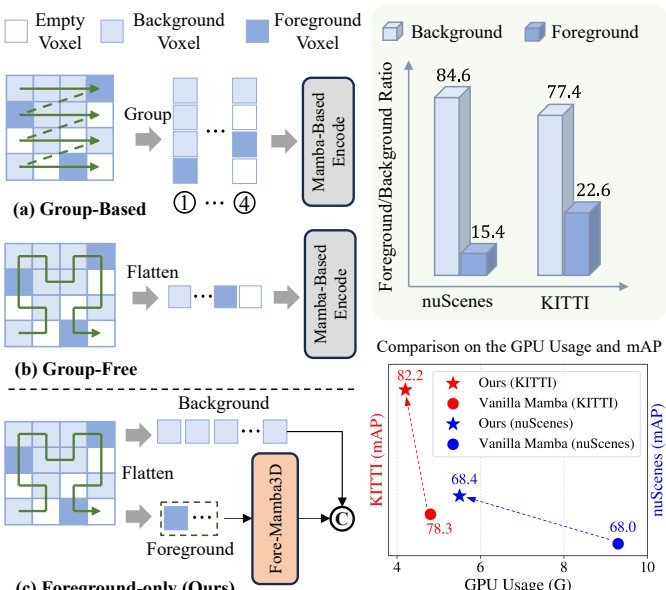

Figure 1: Comparison of previous group-based and group-free Mamba methods with our approach.

semantic and geometric understanding in state variables, which consists of semantic-assisted fusion (SAF) and state spatial fusion (SSF) modules. As shown on the right of the Figure 1, the proposed combinations reduce the memory usage with promising performance achieved simultaneously. The main contributions of our paper are summarized as follows:

- We propose Fore-Mamba3D model, a novel Mamba-based approach that focuses on the effective linear encoding of foreground features for superior 3D detection performance.

- A regional-to-global sliding window strategy is designed to aggregate and propagate local information to the global sequence to address the deficiency of global interaction in previous autoregression models.

- The SASFMamba component is introduced to leverage both semantic-assisted fusion and selective state spatial fusion in state variables, which achieves non-casual encoding with semantic and geometric understanding.

## 2 RELATED WORK

### 2.1 LIDAR-BASED 3D OBJECT DETECTION

LiDAR is an important sensor for 3D perception due to its precise and intuitive geometric representation of scenes. Most of the LiDAR-based 3D detection methods (Shi et al., 2020a; Qi et al., 2017a;b; Vora et al., 2020; Yan et al., 2018; Chen et al., 2022) can be typically categorized into point-based and voxel-based approaches. Point-based methods (Qi et al., 2017a;b; Shi et al., 2019a) directly take raw points as input, which assemble the regional context through set abstraction, and progressively downsampling for point encoding. The above process in point-based methods is computationally inefficient. In contrast, voxel-based methods (Deng et al., 2021; Zhou & Tuzel, 2018; Chen et al., 2023) first divide point clouds into regular voxel grids, and then extract features via 3D SpCNN (Chen et al., 2023; 2022; Wu et al., 2022) or Transformer encoder (Wang et al., 2022b; Bai et al., 2022; Sheng et al., 2021). 3D SpCNN-based methods extract the voxel features by normal convolution kernels, limiting the capability to capture the larger or global scene context. The Transformer-based method groups the voxels in sequence and encodes the information in a self-attention or cross-attention mechanism, which is hindered by the quadratic complexity. Therefore, reducing computational complexity while achieving global recognition remains a challenge.

### 2.2 MAMBA FOR 3D DETECTION

Similar to the process in structured 2D images, the 3D points cloud or voxel representation can also be flattened into serials. Most of the 3D Mamba-based methods (Liang et al., 2024; Liu et al., 2024b; Zhang et al., 2024c; Ning et al., 2024) can be categorized into group-based and group-free classes. PointMamba (Liang et al., 2024) is the pioneer in grouping the point cloud by faster points sampling (FPS) and serializing all the non-empty voxels. LION (Liu et al., 2024b) enables sufficient feature interaction in a much larger group than previous Transformer-based methods, such as DSVT (Wang et al., 2023). In contrast, Voxel-Mamba (Zhang et al., 2024c) is a group-free approach, which serializes the whole space of voxels into a single sequence and enhances the spatial proximity of voxels by designing a dual-scale SSM. MambaDETR (Ning et al., 2024) serializes the queries in the current 3D scene and aggregates the sequential in different frames. However, the linear encoding for all the non-empty voxels is unnecessary and time-cost, while the semantic and spatial relation in the state space is lacking. Our approach tackles these limitations by performing foreground-focused encoding to substantially reduce redundancy.

### 2.3 FOREGROUND SAMPLING AND ENCODING

Foreground sampling and encoding have become central components in recent 3D object detection approaches (Zhang et al., 2022; Wang et al., 2022a; Zhang et al., 2023), particularly for LiDAR-based methods. For example, IA-SSD (Zhang et al., 2022) introduces instance-aware downsampling to hierarchically select foreground points, while RBGNet (Wang et al., 2022a) employs foreground-biased sampling to capture more object-surface points and then applies ray-based feature grouping for improved bounding box prediction. DSASA (Zhang et al., 2023) further develops a series of FPS-based strategies to increase foreground coverage while balancing point density across instances. Nevertheless, a persistent challenge lies in maintaining sufficient representation when the sampled foreground points are sparse and spatially scattered. In this work, we mitigate this issue by designing the specific module to alleviates the response attenuation and information loss arising from sparse foreground-only sequences.

## 3 METHOD

As shown in Figure 2(a), the 3D backbone of Fore-Mamba3D consists of four stages and each contains an instance selection block and a downsampling block. The instance selection block in Figure 2(b) is the main component in our method, which includes foreground voxel sampling, a regional-to-global sliding window (RGSW) strategy in Figure 2(c), and a semantic-assisted and state spatial fusion Mamba (SASFMamba) in Figure 2(d).

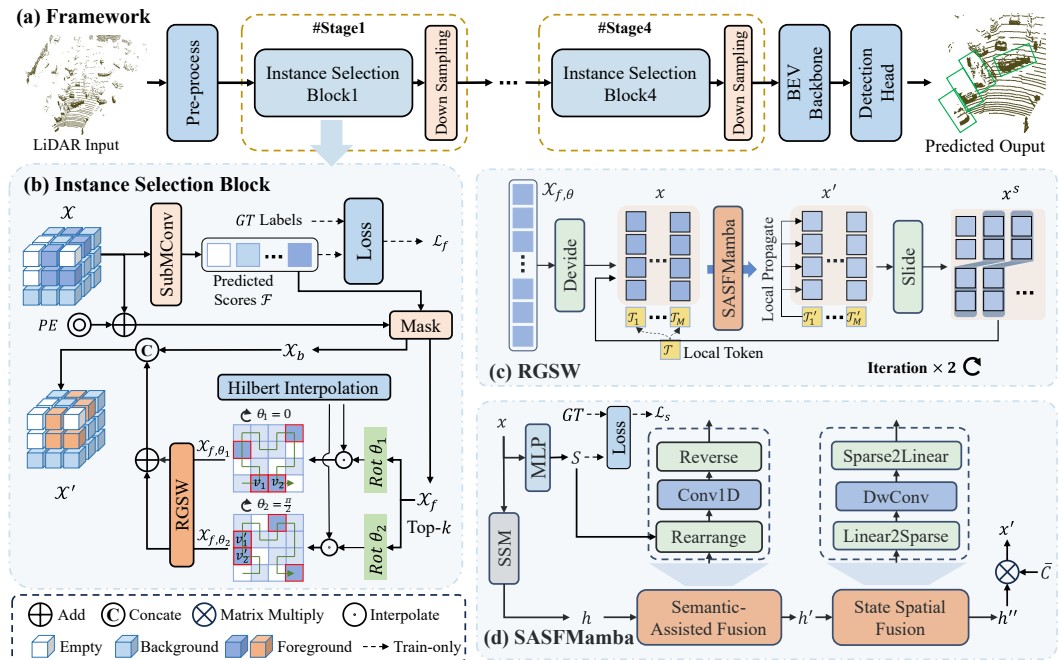

Figure 2: The overall framework of Fore-Mamba3D . (a) Framework: the backbone of Fore-Mamba3D consists of four stages and each contains an instance selection block and a downsampling block. (b) Instance Selection Block: we predict the foreground score for each voxel and select the top-$k$ foreground voxels for further linear encoding. (c) RGSW: we utilize a regional-to-global sliding window process for iterative encoding. (d) SASFMamba: semantic-assisted and state spatial fusion modules are designed to enhance the semantic and geometric recognition of state variables.

## 3.1 FOREGROUND VOXEL SAMPLING AND FLATTENING

In autonomous driving scenarios, background voxels typically account for a substantial portion of the entire scene (e.g., about 80% in the nuScenes or KITTI datasets, as shown in Figure 1). Encoding all nonempty voxels would significantly increase the computational cost and memory usage. To address this challenge, we propose a foreground voxel sampling and encoding method. Given voxel features $\mathcal{X} \in \mathbb{R}^{L \times H \times W \times D}$, where $L \times H \times W$ denotes the spatial resolution and $D$ indicates the channel dimension, we predict foreground score $\mathcal{F}$ for each nonempty voxel via a submanifold convolution. Then, we add the positional embedding $PE$ with $\mathcal{X}$ element-wisely, followed by a sparse convolution operator to obtain the updated features. Subsequently, the top-$k$ ($= \alpha$) updated features are sampled to form our foreground features $\mathcal{X}_f \in \mathbb{R}^{N \times D}$ according to their predicted scores $\mathcal{F}$ in a descending order. The parameter $N$ indicates the number of sampled voxels and the background voxels are defined as $\mathcal{X}_b$. Then, we resort $\mathcal{X}_f$ based on the predefined Hilbert curve.

However, there usually exists 'regional truncation' issue in the Hilbert curve, as illustrated on the bottom of Figure 2(b). Voxels (e.g., $v_1$ and $v_2$) that are close in the original 3D coordinates may be far from each other in the sequence, which cannot be resolved by bidirectional encoding in previous works (Zhu et al., 2024; Zhang et al., 2024c). Therefore, we rotate and flatten the original scene along the Z-axis for multiple times to ensure the truncated neighbor voxels can be closer in the sequence, such as the $v_1'$ and $v_2'$. Specifically, given the grid coordinates of sampled voxels $\mathcal{P} \in \mathbb{R}^{N \times 3}$, we rotate the entire scene with an angle $\theta$ in the bird's eye view (BEV). The initial coordinate $p = (x, y, z)^T$ is transformed to $R(\theta, p) = (\lfloor x cos\theta + y sin\theta \rfloor, \lfloor y cos\theta - x sin\theta \rfloor, z)^T$, where $\lfloor \cdot \rfloor$ indicates the floor function. Subsequently, the voxels are linearized according to the index of their rotated coordinates in the Hilbert curve template. The flattened features can be defined as:

$$\mathcal{X}_{f,\theta} = H(\mathcal{X}_f, \{R(\theta, p) | p \in \mathcal{P}\}) \in \mathbb{R}^{N \times D}, \tag{1}$$

where $\{R(\theta, p) | p \in \mathcal{P}\}$ represents the rotated coordinates set and $H(\cdot)$ refers to the Hilbert curve interpolation function. The sequence features $\mathcal{X}_{f,\theta_i}$ within different $\theta_i$ are fed into the subsequent encoding modules, followed by a sum and multi-layer perception (MLP) operation. The encoded

features are then concatenated with the background features $\mathcal{X}_b$ to produce the final output $\mathcal{X}'$. The above process is formulated as follows:

$$\mathcal{X}' = Cat[MLP(\sum_{i=1}^{r} Enc(\mathcal{X}_{f,\theta_i})), \mathcal{X}_b], \tag{2}$$

where $r$ denotes the rotation times (default as 2). The approach effectively alleviates the regional truncation issue in the Hilbert curve template and improves the robustness of the model across different viewpoints.

## 3.2 REGIONAL-TO-GLOBAL SLIDING WINDOW STRATEGY

To address the response attenuation in the foreground voxels across different instances, we conduct a RGSW strategy for effectively propagating information from the region to the entire sequence, as illustrated in Figure 2(c). We first divide the $N$-length input sequence $\mathcal{X}_{f,\theta}$ into $M$ patches to enable the parallel computation for efficiency. However, due to the causal and auto-regressive nature of Mamba, the model inherently prevents each voxel from accessing later contextual information within the sequence. To compensate for this limitation and eliminate the need for computationally intensive bidirectional encoding, we introduce a local token $\mathcal{T}_i \in \mathbb{R}^D$ inserted into the end of each patch to acquire the expanded feature $x \in \mathbb{R}^{M \times (\frac{N}{M}+1) \times D}$, which is further sent to our SASFMamba model to obtain the encoded feature $x'$. On account of the autoregression property of linear modeling, the encoded local token $\mathcal{T}_i'$ naturally aggregates the comprehensive regional information within the patch $x_i$. Subsequently, we propagate this summarized context $\mathcal{T}_i'$ back to the preceding voxel features within the patch via a similarity-based weighting, defined as:

$$x'_{i,j} = x'_{i,j} + Sim(x'_{i,j}, \mathcal{T}_i') \times \mathcal{T}_i', \tag{3}$$

where $x'_{i,j} \in \mathbb{R}^D$ denotes the $j$-th vector in the $i$-th patch and $Sim(\cdot)$ refers to the cosine similarity.

Although the above approach ensures regional information within each patch can be captured, interaction between patches remains unaddressed. To facilitate global understanding, we implement a sliding window mechanism to update the patch by combining the later half of $x_i'$ with the former half of $x_{i+1}'$ to obtain the new sliding patch $x_i^s$, formulated as:

$$x_i^s = Cat(x_i'[\frac{N}{2M} :], x_{i+1}'[: \frac{N}{2M}]), \tag{4}$$

where $\frac{N}{2M}$ represents the middle position in $x_i'$. Given the obtained $x^s$, it is further fed into SASF-Mamba for information propagation. The above process is repeated for $t$ times to enable information propagation across patches. The technique enables information propagation across patches, which is a critical limitation prevalent in the previous group-based approaches.

## 3.3 SASFMAMBA FOR LINEAR ENCODING

To mitigate the limited contextual representation caused by incomplete and imprecise foreground sampling, we incorporate both geometric and spatial cues and design the SASFMamba encoder. This proposed encoder consists of two critical components: semantic-assisted fusion (SAF) and state-spatial fusion (SSF).

### 3.3.1 SEMANTIC-ASSISTED FUSION

As illustrated in Figure 2(d), given the input sequence $x$, we first employ a lightweight MLP to predict the semantic categories $S$. Simultaneously, we feed $x$ into an SSM to obtain the state variables $h$. The relationship between $h$ and $x$ in a standard SSM is formulated as:

$$h_i = \sum_{j \le i} \bar{A}_{j:i}^{\times} \bar{B}_j x_j; \quad \bar{A}_{j:i}^{\times} = \prod_{t=j+1}^{i} \bar{A}_t, \tag{5}$$

where $h_i$ and $x_j$ denote the $i$-th state variable and the $j$-th input variable, respectively. $\bar{A}_j$ and $\bar{B}_j$ represent the discretized state-transition and input matrices.

To incorporate semantic guidance, we perform a semantic rearrangement on $h$. Specifically, we reorder the state variables $h$ by grouping them according to their predicted categories $S$, while strictly preserving the original relative order within each category sequence. This process aligns voxels with similar semantics regardless of their original positions. Subsequently, a 1D convolution with an effective receptive field (ERF) is applied to this rearranged sequence to aggregate semantic context, followed by a reverse operation to restore the original voxel order, yielding the updated state variables $h'$.

**Theoretical Analysis:** We provide the following derivation to theoretically justify how SAF facilitates long-range information interaction. According to the above process, the updated state variable $h_i'$ (at original index $i$) can be formulated as a weighted sum of its semantic neighbors:

$$h_i' = \sum_{k \in \mathcal{K}} \alpha_k h_{N_k(i)} = \sum_{k \in \mathcal{K}} \sum_{j=1}^{N_k(i)} \alpha_k \bar{A}_{j:N_k(i)}^{\times} \bar{B}_j x_j, \tag{6}$$

where $\mathcal{K} = \{-K, \cdots, K\}$ denotes the ERF and $K$ refers to kernel size in the convolution operation. Crucially, $N_k(i)$ represents the original index of the feature that is spatially adjacent to $i$ in the rearranged semantic domain. $\alpha_k$ is the learnable convolution weight. By expanding $h_{N_k(i)}$, we calculate the association score $\mathbf{M}_{i,j}$ between the updated state $h_i'$ and the input $x_j$ as:

$$\mathbf{M}_{i,j} = \sum_{k \in \mathcal{K}_{i,j}'} \alpha_k \bar{A}_{j:N_k(i)}^{\times} \bar{B}_j; \quad \mathcal{K}_{i,j}' = \{k \in \mathcal{K} \mid N_k(i) > j\}, \tag{7}$$

where $\mathcal{K}_{i,j}'$ denotes the subset of the kernel where the semantic neighbor's original index $N_k(i)$ appears after the input index $j$. From Equation 7, it is evident that if there exists a valid $k \in \mathcal{K}_{i,j}'$, then the cumulative transition $\bar{A}_{j:N_k(i)}^{\times} \neq 0$. Since $\alpha_k$ and $\bar{B}_j$ are non-zero, this ensures $\mathbf{M}_{i,j} \neq 0$. This theoretically demonstrates that the SAF module enables the current state $h_i'$ to effectively capture information from distant inputs $x_j$ (where $j < N_k(i)$) that share similar semantics, overcoming the locality bias of standard linear encoders.

### 3.3.2 STATE SPATIAL FUSION.

To solve the geometric distortion from 3D to 1D sequence, we propose the SSF module. Given the state variables $h'$ after the SAF module, we first map each variable into the 3D space based on its original coordinate in the feature space to create a new sparse 3D tensor in the state space. Since there are inherent spatial deficiency in the linear modeling, we apply a dimension-wise convolution (He et al., 2024) with a large kernel along different axis to achieve spatial recognition. Then, we flatten the 3D representation back into a sequence $h''$. Thus, the overall design of the state space fusion can be expressed as follows:

$$h'' = \text{S2L}(\text{DwConv}(\text{L2S}(h'))), \tag{8}$$

where "L2S" and "S2L" refer to the transformation from linear features to a sparse 3D tensor and the reverse process, respectively. "DwConv" denotes the dimension-wise convolution operation. Based on the observation equation in SSM, we multiply $h''$ by the dynamic output matrix $\bar{C}$ to acquire the output features $x'$. The mechanism in the SSF is similar to SAF, which can ensure the non-casual and geometrically correlated encoding.

### 3.4 LOSS FUNCTION

To improve the precision of the scores $\mathcal{F}$ in foreground prediction and semantic category $S$ in the SAF module, we design two loss functions $\mathcal{L}_f$ and $\mathcal{L}_s$ for supervision. Moreover, we define voxels within enlarged bounding boxes—obtained by expanding the original boxes by 0.5 m along the X/Y axes and 0.25 m along the Z axis—as foreground, in order to preserve ambiguous boundary information. Considering the number imbalance between the predicted categories, we select the focal loss to calculate $\mathcal{L}_f$ and $\mathcal{L}_s$, which are defined as:

$$\mathcal{L}_f, \mathcal{L}_s = -\sum_{i=1}^{C} \beta_i (1 - p_i)^{\gamma} y_i log(p_i), \tag{9}$$

where $C$ defines the number of output classes. The focusing parameter $\gamma = 2$ is introduced to emphasize hard-to-classify samples. $\beta_i$ and $p_i$ represent the weight and probability of the $i$-th category. $y_i$ indicates whether the ground-truth label matches the $i$-th category. After acquiring $\mathcal{L}_f$ and $\mathcal{L}_s$ from the 3D backbone, we integrate them with the classification loss $\mathcal{L}_{cls}$ and regression loss $\mathcal{L}_{reg}$ from the detection head with weight $w = 2$. $\mathcal{L}_{cls}$ and $\mathcal{L}_{reg}$ are computed with the widely used cross-entropy loss and smooth-L1 loss, respectively (Yan et al., 2018; Shi et al., 2020a). The final loss $\mathcal{L}$ is defined as:

$$\mathcal{L} = w(\mathcal{L}_f + \mathcal{L}_s) + \mathcal{L}_{cls} + \mathcal{L}_{reg}. \tag{10}$$

## 4 EXPERIMENTS

### 4.1 DATASET AND METRICS

**nuScenes dataset.** The nuScenes dataset (Caesar et al., 2020) is a large-scale 3D detection benchmark with a perception range of up to 100 meters. The dataset includes 700 training scenes, 150 validation scenes, and 150 testing scenes. Detection performance is evaluated using the normal metrics: mean average precision (mAP) and the nuScenes detection score (NDS), as previous works (Yin et al., 2021; Bai et al., 2022).

**KITTI dataset.** The KITTI dataset (Geiger et al., 2012) contains 3,712 paired training samples, 3,769 paired validation samples, and 7,518 paired test samples. The standard metrics include 3D and BEV average precision (AP) under 40 recall thresholds (R40), with three different difficulty levels. In this work, we evaluate our method across three major categories with IoU thresholds of 0.7 for cars, and 0.5 for both pedestrians and cyclists, which is the same as previous methods (Shi et al., 2020a; Yan et al., 2018; Zhou & Tuzel, 2018).

**Waymo Open Dataset.** The Waymo dataset (Sun et al., 2020) includes 230k annotated samples and the whole scene covers a large reception range of 150 meters. The evaluation metrics contain the average precision (AP) and its variant by weighted heading accuracy (APH). The detection difficulty of each object is divided into two categories: Level 1 (L1) for objects containing more than five points and Level 2 (L2) for those containing at least one point.

### 4.2 IMPLEMENTATION DETAILS

Our model is trained by the Adam optimizer on eight NVIDIA RTX 4090D GPUs with a cosine learning rate of 3e-3. For the nuScenes dataset, our method builds upon the TransFusion framework (Bai et al., 2022) and it is trained for 36 epochs with 2 batch size. For the KITTI dataset, we replace the 3D backbone of the SECOND network (Yan et al., 2018) with our proposed method and train the model for 50 epochs with a batch size of 4. For the Waymo dataset, our method is employed based on the CenterPoint (Yin et al., 2021) with the channel dimension equal to 64 and it is trained for 24 epochs with 2 batch size. We adopt a similar data augmentation configuration in the training stage as previous works (Chen et al., 2023; Yan et al., 2018; Yin et al., 2021; Bai et al., 2022). Notably, the Hilbert curve templates are generated offline and directly loaded into each stage for both training and inference acceleration.

### 4.3 COMPARISON WITH THE PREVIOUS METHODS

**Performance on the nuScenes dataset.** In Table 1, we compare Fore-Mamba3D with previous methods on both the nuScenes validation and test sets. For a fair comparison, all experiments are conducted without class-balanced ground-truth sampling or model ensembling. Fore-Mamba3D achieves state-of-the-art performance among all existing LiDAR-only approaches.

**Performance on the KITTI dataset.** We compare Fore-Mamba3D with other methods based on various 3D backbone architectures, including MLP (Shi et al., 2019a), SpCNN (Zhou & Tuzel, 2018), Transformer (Wang et al., 2023), and Mamba (Liu et al., 2024b), to highlight the effectiveness of our model's structure. The result in Table 2 shows that our method achieves state-of-the-art performance on the KITTI dataset, yielding an average improvement of 1.7% over the second-best method VoxelMamba (Zhang et al., 2024c).

Table 1: Performances on the nuScenes *validation* and *test* set. 'C.V.', 'Ped.', and 'T.C.' are short for construction vehicle, pedestrian, and traffic cone, respectively. The first and second best results are in **bold** and underlined, respectively. All the results are conducted without class-balance ground-truth sampling (CBGS).

| Method | Present at | mAP | NDS | Car | Truck | Bus | Trailer | C.V. | Ped. | Motor. | Bike | T.C. | Barrier |
|---|---|---|---|---|---|---|---|---|---|---|---|---|---|
| Performance on the *validation* dataset |||||||||||||
| CenterPoint (Yin et al., 2021) | CVPR21 | 59.2 | 66.5 | 84.9 | 57.4 | 70.7 | 38.1 | 16.9 | 85.1 | 59.0 | 42.0 | 69.8 | 68.3 |
| TransFusion-L (Bai et al., 2022) | CVPR22 | 65.5 | 70.1 | 86.9 | 60.8 | 73.1 | 43.4 | 25.2 | 87.5 | 72.9 | 57.3 | 77.2 | 70.3 |
| VoxelNeXt (Chen et al., 2023) | CVPR23 | 64.5 | 70.0 | 84.6 | 53.0 | 64.7 | 55.8 | 28.7 | 85.8 | 73.2 | 45.7 | 79.0 | 74.6 |
| DSVT (Wang et al., 2023) | CVPR23 | 66.4 | 71.1 | 87.4 | 62.6 | 75.9 | 42.1 | 25.3 | 88.2 | 74.8 | 58.7 | 77.9 | 71.0 |
| HEDNet (Zhang et al., 2024b) | NIPS23 | 66.7 | 71.4 | 87.7 | 60.6 | 77.8 | 50.7 | 28.9 | 87.1 | 74.3 | 56.8 | 76.3 | 66.9 |
| SAFDNet (Zhang et al., 2024a) | CVPR24 | 66.3 | 71.0 | 87.6 | 60.8 | 78.0 | 43.5 | 26.6 | 87.8 | 75.5 | 58.0 | 75.0 | 69.7 |
| Voxel-Mamba (Zhang et al., 2024c) | NIPS24 | 67.5 | 71.9 | 87.9 | 62.8 | 76.8 | 45.9 | 24.9 | 89.3 | 77.1 | 58.6 | 80.1 | 71.5 |
| LION (Liu et al., 2024b) | NIPS24 | 68.0 | 72.1 | 87.9 | 64.9 | 77.6 | 44.4 | 28.5 | 89.6 | 75.6 | 59.4 | 80.8 | 71.6 |
| FSHNet (Liu et al., 2025) | CVPR25 | 68.1 | 71.7 | 88.7 | 61.4 | 79.3 | 47.8 | 26.3 | 89.3 | 76.7 | 60.5 | 78.6 | 72.3 |
| Fore-Mamba3D (Ours) | – | **68.4** | **72.3** | 88.4 | 65.2 | 80.3 | 48.0 | 28.2 | 89.3 | 75.7 | 57.7 | 80.0 | 71.2 |
| Performance on the *test* dataset |||||||||||||
| HEDNet (Zhang et al., 2024b) | NIPS24 | 67.7 | 72.0 | 87.1 | 56.5 | 70.4 | 63.5 | 33.6 | 87.9 | 70.4 | 44.8 | 85.1 | 78.1 |
| SAFDNet (Zhang et al., 2024a) | CVPR24 | 68.3 | 72.3 | 87.3 | 57.3 | 68.0 | 63.7 | 37.3 | 89.0 | 71.1 | 44.8 | 84.9 | 79.5 |
| Voxel-Mamba (Zhang et al., 2024c) | NIPS24 | 69.0 | 73.0 | 86.8 | 57.1 | 68.0 | 63.2 | 35.4 | 89.5 | 74.7 | 50.8 | 86.9 | 77.3 |
| LION (Liu et al., 2024b) | NIPS24 | 69.8 | 73.9 | 87.2 | 61.1 | 68.9 | 65.0 | 36.3 | 90.0 | 74.0 | 49.2 | 87.3 | 79.5 |
| Fore-Mamba3D (Ours) | – | **70.1** | **74.0** | 87.1 | 60.6 | 70.5 | 63.9 | 34.3 | 90.2 | 73.7 | 53.1 | 88.6 | 78.6 |

Table 2: Comparison of the performance on the KITTI *validation* set with an average recall of 11. ‡ indicates that the result is reproduced by us. The first and second best results are in **bold** and underlined, respectively.

| Methods | BackBone | Car | | | Pedestrian | | | Cyclists | | |
|---|---|---|---|---|---|---|---|---|---|---|
| | | Hard | Mod | Easy | Hard | Mod | Easy | Hard | Mod | Easy |
| PointPillars (Lang et al., 2019) | MLP | 79.1 | 75.0 | 68.3 | 52.1 | 43.5 | 41.5 | 75.8 | 59.1 | 52.9 |
| IA-SSD (Zhang et al., 2022) | MLP | 88.3 | 80.1 | 75.0 | 46.5 | 39.0 | 35.6 | 78.4 | 61.9 | 55.7 |
| VoxelNet (Zhou & Tuzel, 2018) | SpCNN | 77.5 | 65.1 | 57.7 | 39.5 | 33.7 | 31.5 | 61.2 | 48.4 | 44.4 |
| DSVT (Wang et al., 2023) | Transformer | 87.8 | 77.8 | 76.8 | 66.1 | 59.7 | 55.2 | 83.5 | 66.7 | 63.2 |
| DGT-Det (Ren et al., 2023) | Transformer | 89.6 | 80.6 | 78.8 | – | – | – | 82.1 | 68.9 | 61.0 |
| LION (Liu et al., 2024b) | Mamba | 88.6 | 78.3 | 77.2 | 67.2 | 60.2 | 55.6 | 83.0 | 68.6 | 63.9 |
| VoxelMamba (Zhang et al., 2024c)‡ | Mamba | 89.1 | 80.8 | 78.1 | 66.0 | 59.7 | 53.7 | 84.2 | 69.1 | 64.8 |
| Fore-Mamba3D (Ours) | Mamba | **90.3** | **82.2** | **79.5** | **67.8** | **62.2** | **57.0** | **86.4** | **69.5** | **66.3** |

Table 3: Performance on the Waymo dataset (trained on the 20% training dataset and evaluated on the full validation dataset). † denotes that the results of previous work are implemented by OpenPCDet (OpenPCDet Development Team, 2020). The first and second best results are in **bold** and underlined, respectively.

| Methods | Vehicle | | | | Pedestrian | | | | Cyclist | | | | mAP |
|---|---|---|---|---|---|---|---|---|---|---|---|---|---|
| | AP | | APH | | AP | | APH | | AP | | APH | | |
| | L1 | L2 | L1 | L2 | L1 | L2 | L1 | L2 | L1 | L2 | L1 | L2 | L2 |
| SECOND † (Yan et al., 2018) | 71.0 | 62.6 | 70.3 | 62.0 | 65.2 | 57.2 | 54.2 | 47.5 | 57.1 | 55.0 | 55.6 | 53.5 | 58.3 |
| PV-RCNN † (Shi et al., 2020a) | 75.4 | 67.4 | 74.7 | 66.8 | 72.0 | 63.7 | 61.2 | 54.0 | 65.8 | 63.4 | 64.3 | 61.8 | 64.8 |
| Part-A2 † (Shi et al., 2019b) | 74.7 | 65.8 | 74.1 | 65.3 | 71.7 | 62.5 | 62.2 | 54.1 | 66.5 | 64.1 | 65.2 | 62.8 | 64.1 |
| CenterPoint † (Yin et al., 2021) | 71.3 | 63.2 | 70.8 | 62.7 | 72.1 | 64.3 | 65.5 | 58.2 | 68.7 | 66.1 | 67.4 | 64.9 | 64.5 |
| Voxel-RCNN † (Deng et al., 2021) | 76.1 | 68.2 | 75.7 | 67.7 | 78.2 | 69.3 | 72.0 | 63.6 | 70.8 | 68.3 | 69.7 | 67.2 | 68.6 |
| IA-SSD (Zhang et al., 2022) | 70.5 | 61.6 | 69.7 | 60.8 | 69.4 | 60.3 | 58.5 | 50.7 | 67.7 | 65.0 | 65.3 | 62.7 | 62.3 |
| LION (Liu et al., 2024b) | - | 67.0 | - | 66.6 | - | 75.4 | - | 70.2 | - | 71.9 | - | 71.0 | 71.4 |
| Fore-Mamba3D (Ours) | 76.3 | 67.8 | 75.8 | 67.4 | 82.1 | 75.6 | 75.6 | 70.0 | 72.8 | 72.2 | 71.3 | 70.6 | **71.9** |

**Performance on Waymo dataset.** We further evaluate Fore-Mamba3D on a subset of the Waymo Open Dataset, training with only 20% of the training set and testing on the full validation set. As shown in Table 3, our method achieves competitive results with 71.9% mAP in the L2 level, outperforming the CenterPoint baseline (Yin et al., 2021) by 7.4%. Moreover, all the results in the L1 level surpass those of previous methods, further highlighting the effectiveness of our approach.

## 4.4 ABLATION STUDY

In this section, we conduct a series of ablation study on the nuScenes and KITTI validation sets to evaluate the effectiveness and efficiency of all the proposed components.

**Foreground Sampling and Efficiency.** The results in Table 4 demonstrate the effectiveness of our foreground voxel sampling strategy across different sampling ratios, along with its impact on computational efficiency. Specifically, we rotate the entire scene around the Z-axis with $\theta = 0$ and $\pi/2$, and select the top-$k$ (=$\alpha$) features as the foreground. When $\alpha$ equals to 0.2, our model achieves the best performance, with 72.3 NDS on the nuScenes dataset and 82.2 Car mAP on the KITTI dataset. The performance gain can be attributed to the value of $\alpha$ approximating the true distribution. Moreover, we further compare the efficiency under a single-GPU setting with a batch size of 1. Our approach reduces FLOPs by 43.7% and increases FPS by 23.9% compared with the LION backbone. The FLOPs in Mamba are determined by the sequence length, hidden dimension, selective-scan operators, and causal convolution components. In addition, the supervised foreground scores ensure the alignment between the prediction and the ground-truth. Visualization of the heatmap in the BEV is provided in the Appendix.

**Ablation on the Combination of Different Components.** To clearly demonstrate the effectiveness of each component proposed in our method, we fix the sampling ratio $\alpha$ at 0.2 and progressively integrate individual components into the vanilla Mamba encoder. The results are depicted in Table 5. Employing Hilbert curve flattening with multiple rotations achieves certain detection accuracy. Moreover, the RGSW strategy strengthens long-range interactions in the sequence, leading to further performance gains. The SAF and SSF modules focus on capturing semantic and geometric relationships in the foreground representations, respectively. When both of them are incorporated, our model achieves the final promising result.

**Regional-to-Global Sliding Window.** As depicted in Table 6, we explore the validity of the RGSW strategy in mitigating response attenuation. Without either encoding mechanism, the model exhibits suboptimal performance, achieving only 70.2% mAP. Introducing local token insertion or the sliding-window encoding independently improves detection performance by 0.6% and 0.7%, respectively. Notably, the global sliding-window approach provides more remarkable gains for large objects, such as cars ( +1.2%). Meanwhile, the local token insertion is particularly beneficial for smaller and sparser instances, improving the detection of pedestrians (+0.93%) and cyclists (+0.35%). Furthermore, applying more than two iterations of the sliding strategy yields negligible additional gains, indicating a saturation point. Consequently, we adopt $t = 2$ as the default setting to achieve a trade-off between accuracy and efficiency.

Table 4: Efficiency under different sampling ratios and comparison with LION (Liu et al., 2024b).

| $\alpha$ | KITTI | | | nuScenes | | FLOPs (G)↓ | FPS |
|---|---|---|---|---|---|---|---|
| | Car | Ped. | Cyc. | mAP | NDS | | |
| 0.1 | 81.0 | 60.9 | 68.7 | 67.4 | 71.0 | 22.62 | 70 |
| 0.2 | **82.2** | **62.2** | 69.5 | **68.4** | **72.3** | 26.04 | 67 |
| 0.5 | 81.8 | 61.1 | **70.0** | 68.0 | 71.8 | 38.62 | 58 |
| 1.0 | 81.5 | 61.5 | 69.7 | 67.8 | 71.6 | 52.17 | 50 |
| LION | 78.3 | 60.2 | 68.6 | 68.0 | 72.1 | 46.24 | 52 |

Table 5: Performance on the various combinations of proposed components. 'HF' defines the Hilbert flattening.

| HF | RGSW | SAF | SSF | Car | Ped. | Cyc. |
|---|---|---|---|---|---|---|
| ✓ | | | | 79.4 | 59.2 | 66.0 |
| ✓ | ✓ | | | 80.6 | 60.5 | 66.8 |
| ✓ | ✓ | ✓ | | 81.8 | 61.9 | 67.3 |
| ✓ | ✓ | | ✓ | 81.0 | 61.3 | 68.2 |
| ✓ | ✓ | ✓ | ✓ | **82.6** | **62.2** | **69.5** |

Table 6: Ablation on the regional-to-global sliding window. 'Regional' denotes the insertion of local tokens. 'Global' represents the slide window operation. '$t$' denotes the iteration number.

| Regional | Global | $t$ | Car | Ped. | Cyc. | mAP |
|---|---|---|---|---|---|---|
| | | 1 | 80.97 | 60.94 | 68.66 | 70.2 |
| ✓ | | 1 | 81.56 | 61.87 | 69.01 | 70.8 |
| | ✓ | 2 | 82.12 | 61.54 | 68.91 | 70.9 |
| ✓ | ✓ | 2 | **82.16** | **62.23** | 69.46 | **71.3** |
| ✓ | ✓ | 4 | 81.45 | 61.95 | **69.84** | 71.1 |

Table 7: Comparison of applying different kernel sizes in the SAF module. 'Sim' denotes the similarity score between the predicted matrix $M$ and the label.

| $K$ | mAP | NDS | Mems (G)↓ | FLOPs (G)↓ | Sim↑ |
|---|---|---|---|---|---|
| 3 | 68.1 | 71.8 | 4.9 | 22.6 | 0.27 |
| 5 | 68.3 | 72.1 | 5.1 | 23.9 | 0.45 |
| 7 | **68.4** | 72.3 | 5.5 | 26.0 | 0.64 |
| 15 | 68.2 | **72.4** | 6.4 | 30.4 | 0.68 |

**Effectiveness of the State Variable Fusion.** We also investigate the effectiveness of the state variable fusion in enhancing the contextual understanding. As illustrated in Figure 3, we visualize the association matrix $M$ in vanilla SSM, semantic-assisted strategy with different kernel sizes, and

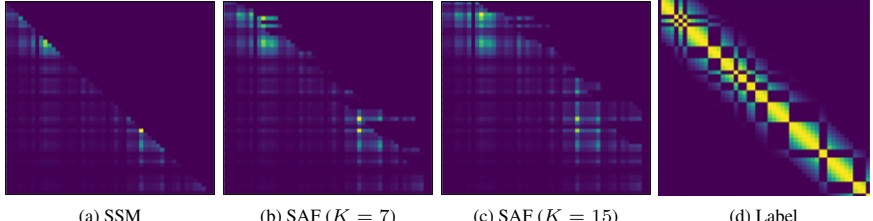

| (a) SSM | (b) SAF ($K = 7$) | (c) SAF ($K = 15$) | (d) Label |

Figure 3: The association matrix in (a) vanilla SSM, (b, c) semantic-assisted strategy with different kernel sizes $K$ and (d) the semantic association labels. We define the semantic labels to follow a Gaussian distribution for the distance.

the semantic association labels. The association matrix $M$ in the vanilla SSM is a lower triangular matrix. After applying our semantic-assisted fusion strategy, the current output can attend to subsequent inputs within similar semantics. Furthermore, as the kernel size $K$ increases, the response in the matrix extends further. This demonstrates that our model could focus on the semantic relationship across the entire sequence. We further compare the detection performance under different kernel sizes in Table 7, which motivates us to adopt a kernel size of 7 as the default setting to balance effectiveness and efficiency.

## 5 CONCLUSION

In this work, we propose Fore-Mamba3D , a novel framework that integrates foreground voxel selection with a hierarchical regional-to-global sliding window strategy to effectively capture inter-instance dependencies in 3D scenes. Furthermore, we introduce the SASFMamba module to enhance both semantic cues and geometric structures, enabling non-causal interactions in the whole linear sequence. Our approach demonstrates substantial performance improvements over existing Mamba-based or foreground-based methods, achieving state-of-the-art results across various autonomous driving benchmarks. Comprehensive ablation studies further validate the effectiveness of each proposed component, highlighting their contribution in advancing the 3D detection task.

ACKNOWLEDGMENTS

This work is partially supported by National Natural Science Foundation of China (Grant No. 62376153, 62402318, 24Z990200676, 62376252), Zhejiang Province Leading Geese Plan(2025C02025, 2025C01056) and Zhejiang Province Province-Land Synergy Program(2025SDXT004-3).

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

APPENDIX

### .1 PERFORMANCE IN DIFFERENT MODEL SCALES

To investigate the impact of the channel dimension in the 3D backbone, we compare the performance of different model scales on the KITTI dataset. As shown in Table 8, the model with a channel dimension of 128 outperforms the model with a channel dimension of 64 by 1.25% and 1.31% mAP in 3D and BEV moderate level detection, respectively.

| Model Scale | 3D | | | BEV | | |
|---|---|---|---|---|---|---|
| | Easy | Mod | Hard | Easy | Mod | Hard |
| -T | 88.53 | 80.91 | 78.46 | 90.87 | 87.78 | 87.27 |
| -B | 90.32 | 82.16 | 79.54 | 92.27 | 89.09 | 88.11 |
| | *(+1.80)* | *(+1.25)* | *(+1.08)* | *(+1.40)* | *(+1.31)* | *(+0.84)* |

Table 8: Performance on different model scales. '-T' indicates the channel dimension in the backbone is 64, and '-B' denotes the channels dimension is set as 128.

### .2 DERIVATION OF STATE VARIABLE FUSION

Defining the initial state $h_0$ in state transition as zero, we rewrite the state transition equation and observation equation in the recursive form as follows:

$$
\begin{aligned}
h_i &= \bar{A}_i h_{i-1} + \bar{B}_i x_i \\
&= \bar{A}_i(\bar{A}_{i-1} h_{i-2} + \bar{B}_{i-1} x_{i-1}) + \bar{B}_i x_i \\
&= \bar{A}_i \bar{A}_{i-1} h_{i-2} + \bar{A}_i \bar{B}_{i-1} x_{i-1} + \bar{B}_i x_i \\
&= \bar{A}_i \cdots \bar{A}_1 h_0 + \bar{A}_i \cdots \bar{A}_2 \bar{B}_1 x_1 + \cdots \\
&\quad + \bar{A}_i \bar{A}_{i-1} \bar{B}_{i-2} x_{i-2} + \bar{A}_i \bar{B}_{i-1} x_{i-1} + \bar{B}_i x_i \\
&= \sum_{j \le i} \left( \prod_{t=j+1}^{i} \bar{A}_t \right) \bar{B}_j x_j = \sum_{j \le i} \bar{A}_{j:i}^{\times} \bar{B}_j x_j; \\
y_i &= C_i h_i + D_i x_i
\end{aligned}
\tag{11}
$$

The output matrix $D_i$ in the observation equation is also omitted, so the output $y_i$ in Equation 6 can be simplified as follows:

$$
y_i = C_i h_i = \sum_{j \le i} C_i \bar{A}_{j:i}^{\times} \bar{B}_j x_j
\tag{12}
$$

Assuming the matrix $\mathbf{M}'$ indicates the association between the output sequence $y = [y_1, y_2, \cdots, y_n]$ and the input sequence $x = [x_1, x_2, \cdots, x_n]$, which is defined by $y_i = \mathbf{M}'_{i,j} x_j$. Combining Equation 7 with the observation equation, $\mathbf{M}'_{i,j}$ can be calculated as:

$$
\mathbf{M}'_{i,j} = \sum_{k \in \mathcal{K}'_{i,j}} \alpha_k \bar{C}_j \bar{A}_{j:N_k(i)}^{\times} \bar{B}_j
\tag{13}
$$

where the $N_k(i)$ could define the semantic or geometric neighbor index. Actually, there usually exists $k$ satisfies $N_k(i) > j$ to ensure $\bar{A}_{j:N_k(i)}^{\times}$ and $\mathbf{M}'_{i,j}$ not equal to 0. Therefore, when the $N_k(i)$ represents the index of a semantically neighboring feature, the output $y_i$ is adaptively recalibrated to prioritize subsequent input features $x_j$ exhibiting high semantic similarity with $x_i$. In contrast, when $N_k(i)$ indicates the index of a geometrically neighboring feature, the output $y_i$ attends to the later input $x_j$ with a spatial proximity to $x_i$.

Moreover, we visualize the process from the $h$ to $h'$ in the SAF module, as shown in the Figure 4. The entire pipeline contains the rearrangement, the 1D convolution operator, and the reverse process. The illustration also corresponds to the transform from Equation 5 to Equation 6.

### .3 VISUALIZATION ON THE FOREGROUND PREDICTION

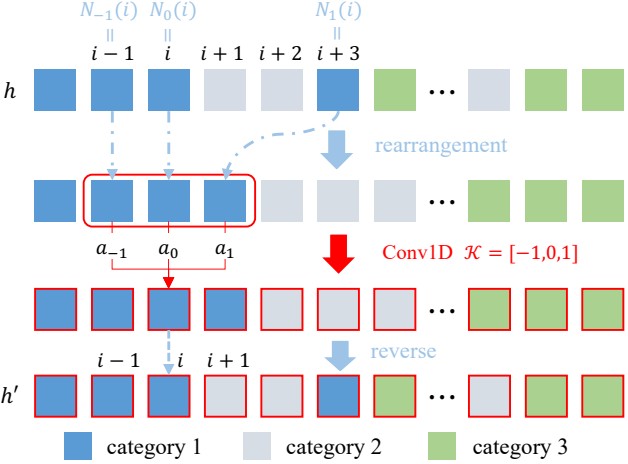

Figure 4: The detailed pipeline in the SAF module, which contains the rearrangement, the 1D convolution, and the reverse process.

We visualize the BEV heatmap of the predicted foreground scores and compare it with the ground truth (GT). Notably, we choose the foreground scores from the third stage in the backbone and accumulate them along the Z-axis. As illustrated in Figure 5, the predicted scores closely match the ground truth values, demonstrating a clear separation between the foreground and background voxels. This demonstrates the high accuracy of our foreground prediction, thereby validating the effectiveness of the proposed foreground sampling strategy.

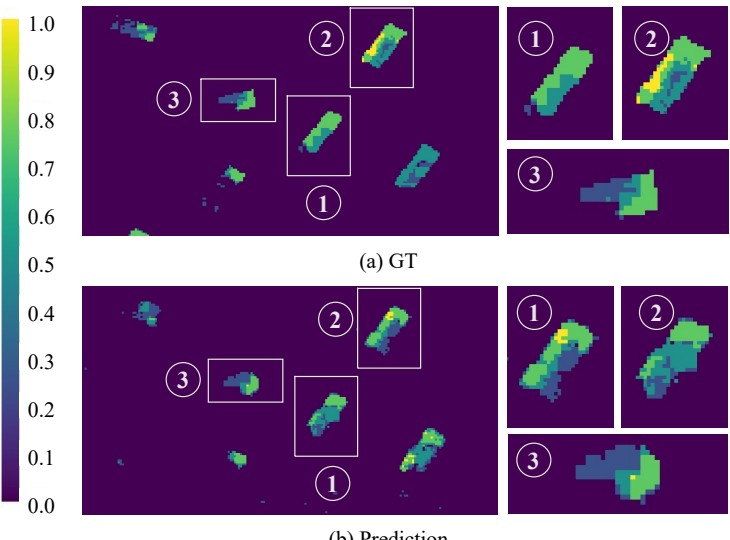

Figure 5: The accuracy in foreground scores prediction. We accumulate the scores along the z-axis and visualize the heatmap in the BEV.

### .4 MORE QUALITATIVE COMPARISON

**Results on the KITTI test dataset.** We also evaluate Fore-Mamba3D on the KITII test dataset. The result in Table 9 shows that our method surpasses the previous work and achieves promising results in LiDAR-only 3D detection.

**Impact of the Rotation and Serialization Strategy.** As shown in Table 10, we freeze $\alpha = 0.2$ and change the rotation angle. The result with $\theta = (0, \pi/2)$ outperforms the result without rotation or with a single rotation, demonstrating the necessity of multiple rotations in flattening for linear encoding. The comparison on the bottom of Table 10 indicates that the simple convolution module outperforms the foreground merging module (Liu et al., 2024b) in the downsampling block. Furthermore, we evaluate the effectiveness of different serialization strategies in Table 11. Compared with the X/Y-raster or the Z-order curve, the Hilbert curve enables the best spatial continuity for precise object detection. Meanwhile, the performance of different Hilbert encoding strategies shows better adaptability to our rotation scheme compared with bidirectional selection.

Table 9: Comparison of the performance on the KITTI *test* dataset with an average recall of 40. All the results are reported in the moderate difficulty level.

| Methods | Car 3D | Car BEV | Ped. 3D | Ped. BEV | Cyc. 3D | Cyc. BEV |
|---|---|---|---|---|---|---|
| PointRCNN (Shi et al., 2019a) | 75.64 | 87.39 | 39.37 | 46.13 | 58.82 | 67.24 |
| PointPillars (Lang et al., 2019) | 74.99 | 86.10 | 43.53 | 50.23 | 59.07 | 62.25 |
| VoxelNet (Zhou & Tuzel, 2018) | 65.11 | 79.26 | 33.69 | 40.74 | 48.36 | 54.76 |
| SECOND (Yan et al., 2018) | 73.66 | - | 42.56 | - | 53.85 | - |
| TANet (Liu et al., 2020) | 75.94 | 86.54 | 44.34 | **51.38** | 59.44 | 63.77 |
| SeSame (Hayeon et al., 2024) | 76.83 | 87.49 | 35.34 | 41.22 | 54.46 | 61.70 |
| Ours | **77.88** | **88.06** | **45.60** | 50.68 | **63.97** | **68.02** |

Table 10: Ablation on the foreground sampling and flattening. 'DS' represents the downsampling ways. '$\alpha$' denotes the sampling ratio and 'Rot. $\theta$' is the rotation angle along the Z-axis.

| DS | $\alpha$ | Rot. $\theta$ | Car | Ped. | Cyc. | mAP |
|---|---|---|---|---|---|---|
| Conv | 0.2 | 0 | 81.31 | 62.16 | 68.23 | 70.6 |
| | 0.2 | $\pi/2$ | 82.01 | 60.55 | 69.17 | 70.6 |
| | 0.2 | $(0, \pi/2)$ | **82.16** | **62.23** | **69.46** | **71.3** |
| Merging | 0.2 | $(0, \pi/2)$ | 81.22 | 61.71 | 66.46 | 69.7 |

Table 11: Performance under different scanning patterns and Hilbert encoding strategies.

| Scan Ways | mAP | NDS | FLOPs (G) ↓ | Latency (ms) ↓ |
|---|---|---|---|---|
| X/Y-raster | 67.4 | 71.5 | 24.5 | 14.6 |
| Z-order | 67.9 | 71.7 | 25.9 | 16.0 |
| Hilbert + Bid | 68.0 | 72.0 | 25.4 | 15.1 |
| Hilbert + Rot | **68.4** | **72.3** | 26.0 | 15.48 |

**Comparison of different sequence modeling approaches.** We also compare the performance on different sequence modeling alternatives, which is shown in Table 12. Specifically, we only replace the sequence encoding part with existing methods (e.g. RetNet Sun et al. (2023), RWKV Peng et al. (2023), and LSTM Hochreiter & Schmidhuber (1997)) and maintain other components consistent with our original pipeline. We train these alternatives from scratch in the nuScenes dataset with the same training configuration. The Mamba model achieves the highest scores in both NDS and mAP metrics. Meanwhile, the computation of Mamba is also efficient compared with the recent RetNet and RWKV approaches. Given the theoretically and empirically validated capability of Mamba in sequence modeling, we adopt it as the default linear encoder throughout our paper.

Table 12: Comparison of the different sequence modeling, including RetNet, RWKV, LSTM, and Mamba.

| Model | NDS | mAP | FLOPs | FPS |
|---|---|---|---|---|
| RetNet | 72.1 | 67.7 | 31.12 | 59 |
| RWKV | 71.9 | 67.1 | 36.15 | 47 |
| LSTM | 70.8 | 65.9 | 23.92 | 78 |
| Mamba | 72.3 | 68.4 | 26.04 | 67 |

**Robustness to the foreground sampling noise.** As shown in the Table 13, our method exhibits robustness to the noise of foreground scoring. Specifically, we directly replace a portion of the sampled foreground voxels (from 5% to 15%) with random background voxels during inference. We then compare the overall detection mAP and NDS results in the nuScenes dataset, as well as the accuracy of specific categories in the nuScenes dataset. It can be observed that our model maintains its performance with a replacement ratio under 10%, which is a relatively large ratio for real-world applications.

Table 13: Comparison of the different ratios of noise in the sampled foreground voxels.

| noise ratio | NDS | mAP | Car | Truck | C.V. | Bus | Trailer | Barrier | Motor | Bike | Ped. | T.C. |
|---|---|---|---|---|---|---|---|---|---|---|---|---|
| w/o | 72.3 | 68.4 | 88.4 | 65.2 | 28.2 | 80.3 | 48.0 | 71.2 | 75.7 | 57.7 | 89.3 | 80.0 |
| 5% | 71.9 | 67.9 | 87.9 | 65.1 | 30.2 | 77.4 | 48.1 | 71.5 | 74.3 | 56.0 | 88.9 | 79.2 |
| 10% | 71.6 | 67.4 | 87.4 | 64.4 | 30.3 | 77.0 | 47.4 | 72.0 | 73.2 | 55.4 | 88.4 | 78.1 |
| 15% | 71.0 | 66.5 | 86.7 | 63.4 | 30.1 | 76.4 | 46.7 | 72.2 | 70.8 | 54.3 | 87.6 | 76.9 |

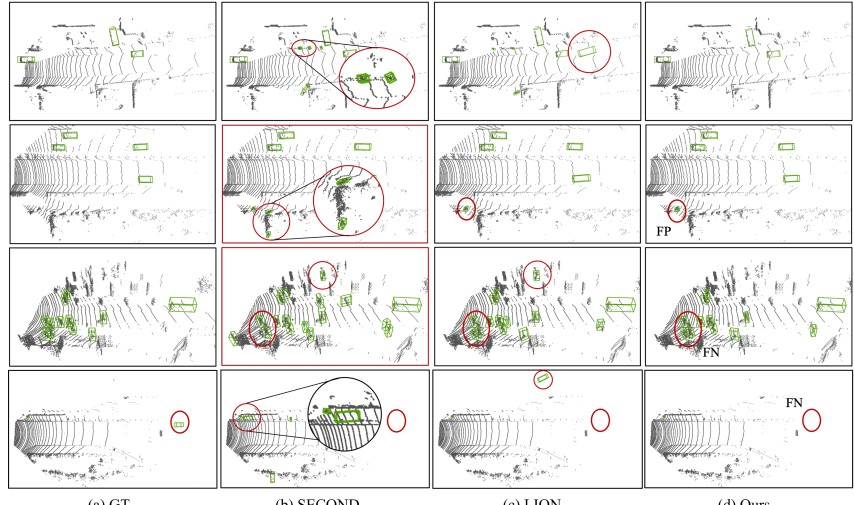

|  (a) GT | (b) SECOND | (c) LION | (d) Ours |

Figure 6: Visualization of the failure cases on the KITTI dataset. Compared with SECOND (Yan et al., 2018) and LION (Liu et al., 2024b), our method can predict the targets more consistently with the ground truth.

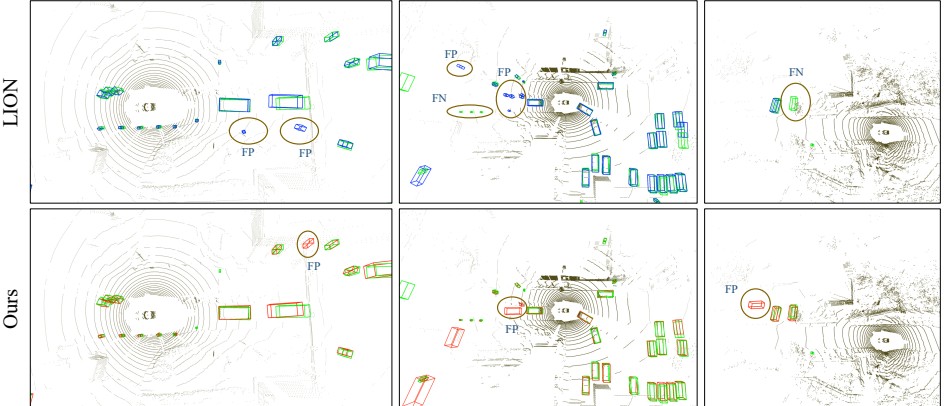

Figure 7: Visualization of the failure cases between our method and the previous method in the nuScenes dataset.

**Visualization of Distinct Cases.** As illustrated in Figure 6, we present a qualitative comparison of the detection results produced by our method and several representative baselines across diverse driving scenarios in the KITTI dataset. The baseline model SECOND (Yan et al., 2018) frequently misclassifies background clutter, such as trees or poles, as pedestrians or cyclists, leading to high false positive rates. Similarly, LION (Liu et al., 2024b) exhibits misdetections in cluttered scenes, often producing false positives when detecting pedestrians and vehicles. In contrast, our proposed method generates predictions that are highly consistent with the ground truth (GT), significantly reducing spurious detections. These visualizations highlight the superior robustness and precision of our approach, especially in challenging cases involving dense and occluded pedestrian instances. We also visualize representative false positive and false negative cases from the nuScenes dataset, as shown in Figure 7. The green boxes denote the ground-truth annotations, while the red and blue boxes represent the predictions of our model and previous methods, respectively. We observe that our approach occasionally produces false negatives on smaller object categories, such as pedestrians and barriers, and certain confusing scenarios may lead to false positives. Nonetheless, the previous method LION also frequently fails on these challenging cases.

