# OpenReview forum: "Fore-Mamba3D: Mamba-based Foreground-Enhanced Encoding for 3D Object Detection"
_ICLR.cc/2026/Conference — ICLR 2026 Poster_

### Official Review · Reviewer_3Dbr · 2025-10-26

**Soundness:** 3
**Presentation:** 2
**Contribution:** 3
**Rating:** 6
**Confidence:** 4

**Summary:**

This paper proposes Fore-Mamba3D, a novel 3D object detection framework that focuses on foreground-enhanced encoding using Mamba-based architectures. The key insight is that previous Mamba-based methods encode entire non-empty voxel sequences containing abundant background information, while foreground voxels occupy only a small portion of the scene. The method introduces two main components: (1) Regional-to-Global Sliding Window (RGSW) strategy to address response attenuation in foreground-only sequences, and (2) Semantic-Assisted and State Spatial Fusion Mamba (SASFMamba) to enrich contextual representation with semantic and geometric awareness.

**Strengths:**

1) The motivation for foreground-focused encoding is compelling and well-supported by empirical analysis.
2) The RGSW strategy effectively addresses the response attenuation problem in foreground-only sequences through a regional-to-global information propagation mechanism.
3) The method achieves state-of-the-art performance across multiple benchmarks (nuScenes: 72.3 NDS, KITTI, Waymo), demonstrating consistent improvements over existing Mamba-based approaches while reducing computational overhead.

**Weaknesses:**

1) While the combination is novel, individual components are relatively incremental. The foreground sampling is essentially top-k selection based on predicted scores, and the Hilbert curve rotation is a straightforward solution to regional truncation issues.
2)  The paper lacks comprehensive ablation studies on key hyperparameters. How sensitive is performance to the number of rotations r,  and the foreground sampling ratio α? The impact of different space-filling curves beyond Hilbert is not explored.
3) The paper doesn't discuss failure cases or limitations of the approach. How does performance vary with different foreground/background ratios across datasets?

**Questions:**

1) How does the method perform when foreground prediction accuracy is low?
2) How sensitive is the method to the choice of space-filling curves (Z-order vs. Hilbert)?

---

> ### Author Response · Authors · 2025-11-24
> **Response to Reviewer 3Dbr Q1**
>
> **1. While the combination is novel, individual components are relatively incremental. The foreground sampling is essentially top-k selection based on predicted scores, and the Hilbert curve rotation is a straightforward solution to regional truncation issues.**
>
> We thank the reviewer for the comment regarding the novelty of individual components. However, our main contribution lies in a foreground-only sequence encoding paradigm for 3D object detection, which is not incremental. Specifically, we target the issues of response attenuation and context deficiency, while simultaneously achieving substantial reductions in computational cost. By designing the foreground sampling, RGSW, and SASF modules in a structured and complementary manner, our approach ensures both efficient computation and robust spatial encoding, going beyond a mere aggregation of incremental techniques.

---

> ### Author Response · Authors · 2025-11-24
> **Response to Reviewer 3Dbr Q2**
>
> **2. The paper lacks comprehensive ablation studies on key hyperparameters. How sensitive is performance to the number of rotations r, and the foreground sampling ratio $\alpha$? The impact of different space-filling curves beyond Hilbert is not explored.**
>
> We thank the reviewer for the comments regarding hyperparameters and space-filling curves. In fact, we have conducted ablations on these hyperparameters in the manuscript and supplementary material in our original submission. 1) We analyze the influence of rotation angle and the number of rotations in Table 10 of the supplementary material. The results show that varying the rotation angle has minimal impact on the final detection mAP, while increasing the number of rotations significantly improves performance (from 70.6 to 71.3). As shown in the lower part of Table 11, the rotated encoding consistently outperforms bidirectional encoding, indicating that multiple rotations effectively mitigate issues at the regional stage. 2) The sensitivity of the sampling ratio $\alpha$ is reported in Table 4 in the paper. When $\alpha$ is set to $0.2$, our model achieves optimal performance on both NuScenes and KITTI datasets. This improvement arises because $\alpha$ closely approximates the true distribution of important points. When $\alpha < 0.2$, insufficient sampling leads to the loss of crucial foreground information; conversely, when $\alpha > 0.2$, excessive background confuses the linear encoding and increases computational cost. 3) We also compare various space-filling curves, including Hilbert, Z-order, and raster scans, as summarized in Table 11 of the supplementary material. Hilbert curves exhibit the strongest spatial continuity and consistently yield the highest detection performance.

---

> ### Author Response · Authors · 2025-11-24
> **Response to Reviewer 3Dbr Q3**
>
> **3. The paper doesn't discuss failure cases or limitations of the approach. How does performance vary with different foreground/background ratios across datasets?**
>
> Thank you for the suggestion to discuss failure cases or limitations of the approach. 1) We have visualized the results of our approach compared with previous works in the KITTI dataset, as illustrated in the Figure 6 of the supplementary materials. It can be observed that the primary negatives and missing detections occur in the small or hard objects, in which previous works also fail to recognize. Moreover, we have also added several failure cases visualization in the nuScenes dataset, as shown in Figure 7. The green boxes denote the ground-truth annotations, while the red and blue boxes represent the predictions of our model and previous methods, respectively. We observe that our approach occasionally produces false negatives on smaller object categories, such as pedestrians and barriers, and certain confusing scenarios may lead to false positives. Nonetheless, the previous method LION also frequently fails on these challenging cases. We have added the visualization of failure modes in the revision (highlighted in lines 920-925 of our revision). 2) We appreciate the reviewer's concern regarding the foreground/background ratios. In our design, we utilize a unified hyperparameter $\alpha$ to adapt to various driving scenarios. Statistics in Figure 1 of the paper show that the foreground occupancy is relatively stable across datasets (15.4% in nuScenes vs. 22.6% in KITTI). The ablation in Table 4 also demonstrates that the model achieves the best performance in both datasets when the number of sampled voxels (=20%) roughly matches the actual foreground voxel count.

---

> ### Author Response · Authors · 2025-11-24
> **Response to Reviewer 3Dbr Q4**
>
> **4. How does the method perform when foreground prediction accuracy is low?**
>
> As reported in the following Table, we directly replace a portion of the sampled foreground voxels with the background voxels during the inference stage, which presents a more disadvantageous situation compared to noisy foreground scoring. We then compare the overall detection mAP and NDS results, as well as the accuracy of specific categories in the nuScenes dataset. It can be observed that our model maintains its performance with a replacement ratio under 10%, which is a relatively large ratio for real-world applications. The result demonstrates the robustness of our model to the inaccurate foreground sampling. We have included this robustness analysis in the supplementary material of the revised version (highlighted in 860-873 lines of our revision).
>
> | noise ratio | NDS | mAP | Car | Truck | C.V. | Bus | Trailer | Barrier | Motor | Bike | Ped. | T.C. |
> | :--- | :---: | :---: | :---: | :---: | :---: | :---: | :---: | :---: | :---: | :---: | :---: | :---: |
> | w/o | 72.3 | 68.4 | 88.4 | 65.2 | 28.2 | 80.3 | 48.0 | 71.2 | 75.7 | 57.7 | 89.3 | 80.0 |
> | 5% | 71.9 | 67.9 | 87.9 | 65.1 | 30.2 | 77.4 | 48.1 | 71.5 | 74.3 | 56.0 | 88.9 | 79.2 |
> | 10% | 71.6 | 67.4 | 87.4 | 64.4 | 30.3 | 77.0 | 47.4 | 72.0 | 73.2 | 55.4 | 88.4 | 78.1 |
> | 15% | 71.0 | 66.5 | 86.7 | 63.4 | 30.1 | 76.4 | 46.7 | 72.2 | 70.8 | 54.3 | 87.6 | 76.9 |

---

> ### Author Response · Authors · 2025-11-24
> **Response to Reviewer 3Dbr Q5**
>
> **5. How sensitive is the method to the choice of space-filling curves (Z-order vs. Hilbert)?**
>
> We have discussed this in Table 11 in our supplementary material of our original submission. In detail, we compare different space-filling curves, including X/Y raster scans, Z-order curves, and our Hilbert curves. X/Y raster scans lead to a notable drop in both mAP and NDS, indicating that arbitrary concatenation severely disrupts spatial consistency. Z-order curves preserve only partial locality and also perform worse. In contrast, our multi-rotated Hilbert ordering consistently achieves the highest mAP and NDS, demonstrating superior spatial coherence. These results confirm that our choice of ordering is not arbitrary, but empirically validated as the most effective and robust strategy.

---

### Official Review · Reviewer_jtLi · 2025-10-28

**Soundness:** 3
**Presentation:** 2
**Contribution:** 3
**Rating:** 4
**Confidence:** 4

**Summary:**

Fore-Mamba3D is a Mamba-based 3D perception backbone for LiDAR-based object detection. It focuses on foreground voxels predicted by a learned scoring function. It propose a Regional-to-Global Sliding Window module which enhances long-range voxel interaction to prevent information loss from sparse foregrounds. It also introduce a Semantic-Assisted State Fusion module for fusing semantic and geometric features. It achieves efficient linear modeling and how state-of-the-art performance on nuScenes, KITTI, and Waymo.

**Strengths:**

1. The paper presents an interesting and effective idea by selectively adapting foreground voxels for Mamba-based modeling. This approach substantially reduces computational overhead while maintaining high representational quality, showing a smart trade-off between efficiency and performance.

2. The method is validated across multiple benchmark datasets (nuScenes, KITTI, Waymo) with consistent performance gains. The visual results Fig3 are clear and insightful, effectively illustrating how the proposed model can improve the contextual understanding of fused voxel features.

**Weaknesses:**

1. Although the proposed method introduces innovative foreground-focused encoding, the overall performance gain compared to prior Mamba-based or Transformer-based 3D detectors is relatively modest. The improvements, while consistent, may not fully justify the added architectural complexity or the additional design components.

2. Unclear Spatial Consistency in the Regional-to-Global Sliding Window (RGSW): In the proposed RGSW strategy, the authors directly fuse the later half of the sequence with the previous half, which raises concerns about spatial coherence. Since the voxel sequence is constructed by selectively sampled voxels along a Hilbert curve, their ordering does not guarantee true spatial adjacency. This makes the fusion operation confusing, the merged regions may not correspond to physically neighboring spatial areas. As a result, the process may not fully respect local spatial correlation, and in extreme cases, it could behave similarly to random voxel concatenation, weakening the interpretability of the regional-to-global transition.

3. The authors completely discard background voxel feature extraction, directly concatenating the processed foreground features. While this design choice further reduces computational cost, it introduces a potential issue, semantic strength inconsistency between foreground and background regions. In fact, background voxels play an important role in defining object boundaries and separating adjacent instances. Ignoring background information entirely may lead to inaccurate boundary localization, especially in cluttered or occluded scenes.

**Questions:**

Have the authors evaluated how alternative sequence ordering strategies, such as random shuffling or different spatial traversal patterns, affect the performance of the Regional-to-Global Sliding Window (RGSW)? If so, how sensitive is the model to the chosen voxel sequence order, and does it significantly influence spatial consistency or detection accuracy?

---

> ### Author Response · Authors · 2025-11-24
> **Response to Reviewer jtLi Q1**
>
> **1. Although the proposed method introduces innovative foreground-focused encoding, the overall performance gain compared to prior Mamba-based or Transformer-based 3D detectors is relatively modest. The improvements, while consistent, may not fully justify the added architectural complexity or the additional design components.**
>
> We appreciate that you recognize the novelty in our paper. However, our method is the first to employ a foreground-only sequence encoding paradigm for 3D object detection, and it surpasses previous global-encoding approaches in both accuracy and efficiency. Since our framework operates in a foreground-only manner, additional pipeline optimization and design proposed in our paper is essential and necessary. Specifically, we target the issues of response attenuation and context deficiency, while simultaneously achieving substantial reductions in computational cost. As shown in Table 1 in our paper, our method achieves 68.4 mAP and 72.3 NDS on the nuScenes dataset, outperforming strong state-of-the-art approaches such as LION and Voxel-Mamba. At the same time, our framework reduces computation by 43.7% FLOPs and improves runtime efficiency by 23.9% FPS in the encoder, as reported in Table 4 in the paper. This combination of higher accuracy together with significantly lower computational overhead has not been achieved in prior work. Moreover, on the KITTI and Waymo datasets (Table 2 and Table 3), our method continues to deliver consistent improvements of +1.4% (Car Moderate mAP on KITTI) and +0.5% (Waymo mAP L2). In the 3D object detection community, such improvements are typically considered indicative of substantial architectural or representational advances. For example, Voxel-Mamba improved NDS by 0.5% (from 71.4% to 71.9%) over prior arts, and FSHNet achieved a gain of 0.6% (from 71.1% to 71.7%) according to their own papers. Therefore, the improved results and gain in efficiency demonstrate that demonstrates that our method provides more than incremental improvements and introduces a meaningful advancement to 3D detection.

---

> ### Author Response · Authors · 2025-11-24
> **Response to Reviewer jtLi Q2**
>
> **2. Unclear Spatial Consistency in the Regional-to-Global Sliding Window (RGSW): In the proposed RGSW strategy, the authors directly fuse the later half of the sequence with the previous half, which raises concerns about spatial coherence. Since the voxel sequence is constructed by selectively sampled voxels along a Hilbert curve, their ordering does not guarantee true spatial adjacency. This makes the fusion operation confusing, the merged regions may not correspond to physically neighboring spatial areas. As a result, the process may not fully respect local spatial correlation, and in extreme cases, it could behave similarly to random voxel concatenation, weakening the interpretability of the regional-to-global transition.**
>
> Thanks for the concern regarding the design of the RGSW module. Notably, RGSW does not compromise the spatial continuity of local regions. This is because the unfolding is performed along the Hilbert curve, which preserves spatial proximity throughout the sequence. Consequently, the latter half of one patch and the former half of the next patch remain adjacent in 3D space, and concatenating them preserves the original spatial ordering and locality. Moreover, each patch has a sufficiently long sequence—e.g., 4,096 voxels in the first instance selection block—covering numerous foreground instances. Sampling from the middle portion of such a long sequence does not disrupt the continuity of the remaining instances. Overall, RGSW can be viewed as a composition of multiple group-based encoders whose spatial coherence and detection effectiveness have been validated in prior works such as DSVT and Voxel-Mamba. The key difference is that our sliding-window partitioning introduces cross-group interactions, effectively alleviating the information-blocking problem in previous methods and substantially improving the model’s ability to capture global context.

---

> ### Author Response · Authors · 2025-11-24
> **Response to Reviewer jtLi Q3**
>
> **3. The authors completely discard background voxel feature extraction, directly concatenating the processed foreground features. While this design choice further reduces computational cost, it introduces a potential issue, semantic strength inconsistency between foreground and background regions. In fact, background voxels play an important role in defining object boundaries and separating adjacent instances. Ignoring background information entirely may lead to inaccurate boundary localization, especially in cluttered or occluded scenes.**
>
> Thank you for raising this thoughtful concern regarding the potential loss of background context. However, our method does not completely discard background information. Although the instance selection block primarily focuses on foreground points, the down-sampling block still encodes all non-empty voxels, thereby preserving both foreground and background features. More importantly, in the instance selection block, foreground voxels are defined using an enlarged grounding box rather than the original bounding box. This box is expanded to ensure that the selected voxels naturally include ambiguous boundary regions instead of only interior object points, which helps preserve informative border cues and omits the background information that is far away from the border. Please also refer to the reply to the 2nd question of Reviewer cXEa for more details. We have clarified the implementation of enlarged boxes in section 3.4 in the revision (highlighted in 303-306 lines of our revision).

---

> ### Author Response · Authors · 2025-11-24
> **Response to Reviewer jtLi Q4**
>
> **4. Have the authors evaluated how alternative sequence ordering strategies, such as random shuffling or different spatial traversal patterns, affect the performance of the Regional-to-Global Sliding Window (RGSW)? If so, how sensitive is the model to the chosen voxel sequence order, and does it significantly influence spatial consistency or detection accuracy?**
>
> Thank you for raising the concern regarding the impact of sequence ordering strategies. In fact, we have conducted an ablation study on the sensitivity to sequence ordering, as reported in Table 11 of the supplementary material in our original submission. The results show that the rotated Hilbert curve used in our method is the most stable and spatially coherent ordering strategy. We compare several alternatives, including X/Y raster scans, Z-order curves, and single-angle or bidirectional Hilbert curves. X/Y raster scans lead to a notable 1.0% mAP drop, indicating that arbitrary concatenation severely disrupts spatial consistency. Z-order and single-angle Hilbert curves retain only partial locality and perform worse. In contrast, the rotated Hilbert ordering consistently achieves the highest mAP and NDS, demonstrating both superior spatial coherence and the best compatibility with RGSW. These findings confirm that our ordering choice is not arbitrary but empirically validated as the most effective and robust strategy. They also show that RGSW specifically benefits from a coherent spatial sequence rather than any incidental sequence property.

---

### Official Review · Reviewer_7P9E · 2025-10-29

**Soundness:** 3
**Presentation:** 3
**Contribution:** 3
**Rating:** 6
**Confidence:** 4

**Summary:**

3D object detection is a critical task in computer vision with broad applications in autonomous driving. In this paper, the authors proposed a novel LiDAR based 3D object detector, named as FORE-MAMBA3D. It demonstrates the better performance in the KITTI, NuScenes, and Waymo datasets.

**Strengths:**

This paper is well-structured and supported by sufficient experiments, showing great potential for acceptance.
1. Experiments are conducted in the mainstreams datasets, such as KITTI, NuScenes, and Waymo datasets. These results show the effectinvess of the proposed method.
2. Overview of the proposed framework is very clear. It shows the novelty of the proposed method.
3. Motivation of the proposed method is also clear.

**Weaknesses:**

The following points are suggested for further improvement.
​​Q1. In the Introduction, the mention of "incomplete and imprecise sampling" seems disconnected from the preceding context, which might confuse readers about the purpose of this sampling operation. Please clarify its objective. For instance, if foreground-only encoding is used, does it require sampling foreground voxels? Please explain this clearly.
​​Q2. In the Related Work section, the authors highlight the limitations of existing methods. It would be better to further emphasize how the proposed method in this paper specifically addresses these limitations.
​​Q3. In Section 3.2, the authors introduce a local token to summarize information within each patch. It is recommended that the authors explain (1) their motivation (e.g., to address response attenuation) (2) explain  the need for a local token to re-summarize information, (3) finally delve into the specific computational details. Currently, the local token is introduced somewhat abruptly.
​​Q4. Section 3.3.1 does not clearly explain the computational process and the motivation behind the proof, even with supplementary materials. The authors need to clarify:
a. How exactly are voxels rearranged using the semantic categories S? Is cosine similarity used for this rearrangement?
b. From the diagram, it appears that x undergoes only one SSM operation and one 1D convolution to obtain h'. However, Equation 6 seems confusing. For example, is Nk(i) "the original index of the semantically neighboring feature" (as in Section 3.2) or the "index of semantically neighboring feature" (as in the supplementary material)? Furthermore, in Equation 6, is index j the index before or after rearrangement? Also, does merely changing the order allow the transformation from Equation 5 to Equation 6, given that the SSM operation is performed on the original sequence?
c. It would be beneficial to separate the description of the network's computational process from the theoretical proof in Section 3.2. At the very least, indicate to readers that the subsequent formulas are primarily for theoretical justification and are not necessary during network inference.
​​Q5. Please specify under what configuration (one 4090D? 1batch size?) the network's inference speed was evaluated in the experiments. Furthermore, I am curious about how the FLOPs for the Mamba module within the 3D backbone were computed.

**Questions:**

The following points are suggested for further improvement.
​​Q1. In the Introduction, the mention of "incomplete and imprecise sampling" seems disconnected from the preceding context, which might confuse readers about the purpose of this sampling operation. Please clarify its objective. For instance, if foreground-only encoding is used, does it require sampling foreground voxels? Please explain this clearly.
​​Q2. In the Related Work section, the authors highlight the limitations of existing methods. It would be better to further emphasize how the proposed method in this paper specifically addresses these limitations.
​​Q3. In Section 3.2, the authors introduce a local token to summarize information within each patch. It is recommended that the authors explain (1) their motivation (e.g., to address response attenuation) (2) explain  the need for a local token to re-summarize information, (3) finally delve into the specific computational details. Currently, the local token is introduced somewhat abruptly.
​​Q4. Section 3.3.1 does not clearly explain the computational process and the motivation behind the proof, even with supplementary materials. The authors need to clarify:
a. How exactly are voxels rearranged using the semantic categories S? Is cosine similarity used for this rearrangement?
b. From the diagram, it appears that x undergoes only one SSM operation and one 1D convolution to obtain h'. However, Equation 6 seems confusing. For example, is Nk(i) "the original index of the semantically neighboring feature" (as in Section 3.2) or the "index of semantically neighboring feature" (as in the supplementary material)? Furthermore, in Equation 6, is index j the index before or after rearrangement? Also, does merely changing the order allow the transformation from Equation 5 to Equation 6, given that the SSM operation is performed on the original sequence?
c. It would be beneficial to separate the description of the network's computational process from the theoretical proof in Section 3.2. At the very least, indicate to readers that the subsequent formulas are primarily for theoretical justification and are not necessary during network inference.
​​Q5. Please specify under what configuration (one 4090D? 1batch size?) the network's inference speed was evaluated in the experiments. Furthermore, I am curious about how the FLOPs for the Mamba module within the 3D backbone were computed.

---

> ### Author Response · Authors · 2025-11-24
> **Response to Reviewer 7P9E Q1**
>
> **1. In the Introduction, the mention of "incomplete and imprecise sampling" seems disconnected from the preceding context, which might confuse readers about the purpose of this sampling operation. Please clarify its objective. For instance, if foreground-only encoding is used, does it require sampling foreground voxels? Please explain this clearly.**
>
> We thank the reviewer for the insightful suggestion. Our method relies on foreground-only encoding for 3D object detection, which inherently requires accurate prediction and sampling of foreground voxels. Hence, incomplete and imprecise sampling may omit critical foreground structures, leading to degraded detection performance. To mitigate this issue, we predict a foreground score for each voxel and perform sampling in descending order of these scores. We have clarified this motivation and the role of the sampling mechanism more explicitly in the modified Introduction (highlighted in 52-54 lines of our revision).

---

> ### Author Response · Authors · 2025-11-24
> **Response to Reviewer 7P9E Q2**
>
> **2. In the Related Work section, the authors highlight the limitations of existing methods. It would be better to further emphasize how the proposed method in this paper specifically addresses these limitations.**
>
> Thank you for the valuable suggestion regarding related work. Existing sequence-based detectors process all non-empty voxels globally, which makes them inefficient and susceptible to redundancy introduced by large background regions. Our method addresses these issues by (1) performing foreground-focused encoding to substantially reduce redundancy, and (2) introducing the RGSW and SASF modules to alleviate response attenuation and information loss arising from foreground-only sequences. We have strengthened the discussion of these advantages in the revised related work section (highlighted in 126-127 and lines 138-140 of our revision).

---

> ### Author Response · Authors · 2025-11-24
> **Response to Reviewer 7P9E Q3**
>
> **3. In Section 3.2, the authors introduce a local token to summarize information within each patch. It is recommended that the authors explain (1) their motivation (e.g., to address response attenuation) (2) explain the need for a local token to re-summarize information, (3) finally delve into the specific computational details. Currently, the local token is introduced somewhat abruptly.**
>
> We appreciate the reviewer’s feedback on the motivation and details of our local token in the RGSW module. We clarify that 1) Motivation: Due to the causal and auto-regressive nature of Mamba, the model prevents each voxel from accessing later contextual information and compensates for response attenuation. 2) Necessity: Therefore, we introduce the local token to capture global information in the current patch sequence, which eliminates the utilization of bidirectional encoding and enables effective global information interaction in the following patch sliding. 3) Computational details: Specifically, after grouping the entire foreground voxels into several patches, we append a local token at the end of each patch and conduct the auto-regressive encoding to aggregate within-patch whole information. Then, we propagate this information to preceding voxels via cosine-similarity-based weighting. We have expanded the motivation and explanation in the revision (highlighted in 217-220 lines of our revision).

---

> ### Author Response · Authors · 2025-11-24
> **Response to Reviewer 7P9E Q4**
>
> **4. Section 3.3.1 does not clearly explain the computational process and the motivation behind the proof, even with supplementary materials. The authors need to clarify: a. How exactly are voxels rearranged using the semantic categories S? Is cosine similarity used for this rearrangement? b. From the diagram, it appears that x undergoes only one SSM operation and one 1D convolution to obtain h'. However, Equation 6 seems confusing. For example, is Nk(i) "the original index of the semantically neighboring feature" (as in Section 3.2) or the "index of semantically neighboring feature" (as in the supplementary material)? Furthermore, in Equation 6, is index j the index before or after rearrangement? Also, does merely changing the order allow the transformation from Equation 5 to Equation 6, given that the SSM operation is performed on the original sequence? c. It would be beneficial to separate the description of the network's computational process from the theoretical proof in Section 3.2. At the very least, indicate to readers that the subsequent formulas are primarily for theoretical justification and are not necessary during network inference.**
>
> Thanks for the suggestion on Section 3.3.1. a) We first predict the semantic category of each voxel using a lightweight MLP. Voxels are then re-arranged by aggregating them according to predicted categories while preserving their original order within each category sequence, which is defined as the ``re-arrangement'' stage in the Figure 4 in the supplementary material of the revision. Notably, cosine similarity is not used for re-arrangement. b) From the diagram, $x$ undergos an SSM operation to obtain $h$. Then, the re-arrangement, the 1D convolution, and the reverse process are further conducted to obtain $h'$, which is illustrated in Figure 4. In Equation 6, $i$, $N_k(i)$, and $j$ all denote the original sequence indices in $h$. $N_k(i)$ refers to the original index of the features that are semantically adjacent to the $i$-th voxel. Moreover, the transition from Equation 5 to 6 reflects both the semantic rearrangement and the subsequent 1D convolution operation. Semantic arrangement allows spatially distant items to be aggregated together, while the 1D convolution enables effective interaction among them. c) In the revision, we have rephrased Section 3.3.1, in which we separate the description of the computational pipeline from the theoretical justification and explicitly state that the mathematical derivations are for theoretical analysis.

---

> ### Author Response · Authors · 2025-11-24
> **Response to Reviewer 7P9E Q5**
>
> **5. Please specify under what configuration (one 4090D? 1batch size?) the network's inference speed was evaluated in the experiments. Furthermore, I am curious about how the FLOPs for the Mamba module within the 3D backbone were computed.**
>
> The reported inference speed is evaluated on a single RTX 4090D GPU with a batch size of 1. For FLOPs estimation of the Mamba module within our 3D backbone, we follow the counting method used in the prior SSM-based approach LION, computing FLOPs based on the sequence length, hidden dimension, selective scan operators, and causal convolutional components. We have added these details in the corresponding ablation study (highlighted in 427-430 lines of our revision).

---

### Official Review · Reviewer_cXEa · 2025-10-31

**Soundness:** 3
**Presentation:** 3
**Contribution:** 3
**Rating:** 6
**Confidence:** 3

**Summary:**

This paper introduces a Mamba-based backbone for 3D object detection focused on more effective foreground voxel encoding. The method selects top-scoring foreground voxels, serializes them using a rotation-augmented Hilbert curve, and processes them with RGSW for propagating local and global context, and SASFMamba to enhance context representation via state space modeling. Experiments are performed on nuScenes, KITTI, and Waymo benchmarks, along with ablation studies to justify each component.

**Strengths:**

1. The paper clearly identifies a major inefficiency in prior Mamba-based 3D object detectors: unnecessary computation over background voxels, and the resulting context loss when naïvely restricting to the foreground.
2. Foreground selection is performed using a trainable scoring mechanism with effective sampling and serialization, alleviating regional truncation problems. It is well-motivated both by experiments and design illustrations.
3. The RGSW and the SASFMamba fusion modules are described in substantial detail, with concrete mathematical rationale, and the implementation roadmap is well clarified.
4. The method is compared against a rich spectrum of modern baselines on three benchmarks. Relative improvements are numerically strong.

**Weaknesses:**

1. Limited discussion of causal sequence modeling approaches: Although the method addresses linear sequence encoding with foreground focus, it omits explicit comparison to other causal sequence modeling alternatives or other advanced state-space models.
2. Modeling limitations: The method largely treats background voxels as non-informative, but in autonomous driving, background structure or clutter can contain subtle contextual cues (for object contextual priors, occlusion estimation, etc.). The effect of completely de-emphasizing background is not discussed in terms of potential blindspots for categories with ambiguous boundaries.

**Questions:**

1. How robust is the method to imperfect, noisy, or ambiguous foreground scoring? Do detection performance or ablations degrade gracefully if the scoring network fails to highlight the correct voxels?
2. How do the authors envision handling the omission of background information in domains where background context could be a valuable cue (e.g., occlusions, instance support)?
3. Please elaborate on any observed failure modes (missed detection, false positives/negatives), especially with reference to dataset bias, rare classes, or environmental conditions.

---

> ### Author Response · Authors · 2025-11-24
> **Response to Reviewer cXEa Q1**
>
> **1. Limited discussion of causal sequence modeling approaches: Although the method addresses linear sequence encoding with foreground focus, it omits explicit comparison to other causal sequence modeling alternatives or other advanced state-space models.**
>
> We appreciate the reviewer’s suggestion regarding the performance of diverse causal sequence modeling approaches. The results in Tables 1, 2, and 3 in the paper demonstrate the effectiveness of our approach compared with previous Mamba-based 3D detectors, such as LION and Voxel-Mamba. Moreover, we also compare the performance of the Mamba model with other alternatives, which is shown in the following Table. Specifically, we only replace the sequence encoding part with existing methods (e.g. RetNet (Ref.1), RWKV (Ref.2), and LSTM (Ref.3)) and maintain other components consistent with our original pipeline. We train all these alternatives from scratch in the nuScenes dataset with the same training configuration. The results prove that the Mamba model achieves the highest scores in both NDS and mAP metrics. Meanwhile, the computation of Mamba is also more efficient compared with the recent RetNet and RWKV approaches. Given the theoretically and empirically validated capability of Mamba in sequence modeling, we adopt it as the default linear encoder throughout our paper. We have also included this comparison in Table 12 of the supplementary material in the revised version (highlighted in 842-858 lines of our revision).
>
> | Model | NDS | mAP | FLOPs (G) $\downarrow$ | FPS |
> | :--- | :---: | :---: | :---: | :---: |
> | RetNet | 72.1 | 67.7 | 31.12 | 59 |
> | RWKV | 71.9 | 67.1 | 36.15 | 47 |
> | LSTM | 70.8 | 65.9 | 23.92 | 78 |
> | Mamba (Ours) | 72.3 | 68.4 | 26.04 | 67 |
>
> Ref.1: Y. Sun, L. Dong, S. Huang, S. Ma, Y. Xia, J. Xue, J. Wang, and F. Wei, “Retentive network: A successor to transformer for large language models,” arXiv preprint arXiv:2307.08621, 2023.
>
> Ref.2: B. Peng, E. Alcaide, Q. Anthony, A. Albalak, S. Arcadinho, S. Biderman, H. Cao, X. Cheng, M. Chung, M. Grella et al., “Rwkv: Reinventing rnns for the transformer era,” arXiv preprint arXiv:2305.13048, 2023.
>
> Ref.3: S. Hochreiter and J. Schmidhuber, “Long short-term memory,” Neural computation, vol. 9, no. 8, pp. 1735–1780, 1997.

---

> ### Author Response · Authors · 2025-11-24
> **Response to Reviewer cXEa Q2**
>
> **2. Modeling limitations: The method largely treats background voxels as non-informative, but in autonomous driving, background structure or clutter can contain subtle contextual cues (for object contextual priors, occlusion estimation, etc.). The effect of completely de-emphasizing background is not discussed in terms of potential blindspots for categories with ambiguous boundaries.**
>
> We appreciate the opportunity to clarify our handling of background information. It is worth noting that our method does not completely de-emphasize background information. Though the instance selection block focuses on foreground encoding, the down-sampling block still encodes all non-empty voxels, ensuring that global background representations are retained. More importantly, in the implementation of the instance selection block, we define foreground voxels as those falling inside an enlarged grounding box rather than the original one. Specifically, the box is expanded by 0.5 m along the X/Y axes and 0.25 m along the Z axis, which intentionally includes ambiguous boundary regions instead of only strict object-interior voxels. This design ensures that the sampled foreground preserves informative border cues. For background voxels that lie farther from the original boxes, we argue that such distant context offers little meaningful contribution to accurate box localization. Therefore, our design deliberately preserves near-boundary contextual cues while avoiding the unnecessary inclusion of irrelevant background. We have clarified the implementation of enlarged boxes in section 3.4 in the revision (highlighted in 303-306 lines of our revision).

---

> ### Author Response · Authors · 2025-11-24
> **Response to Reviewer cXEa Q3**
>
> **3. How robust is the method to imperfect, noisy, or ambiguous foreground scoring? Do detection performance or ablations degrade gracefully if the scoring network fails to highlight the correct voxels?**
>
> As shown in the following Table, our method exhibits robustness to the noise of foreground scoring. Specifically, a portion of the sampled foreground voxels (from 5% to 15%) is directly replaced with the background voxels during inference, which presents a more disadvantageous situation compared to noisy foreground scoring. We then compare the overall detection mAP and NDS results, as well as the accuracy of specific categories in the nuScenes dataset. It can be observed that our model maintains its performance with a replacement ratio under 10%, which is a relatively large ratio for real-world applications. We have included this robustness analysis in the supplementary material of the revised version (highlighted in 860-873 lines of our revision).
>
> | noise ratio | NDS | mAP | Car | Truck | C.V. | Bus | Trailer | Barrier | Motor | Bike | Ped. | T.C. |
> | :--- | :---: | :---: | :---: | :---: | :---: | :---: | :---: | :---: | :---: | :---: | :---: | :---: |
> | w/o | 72.3 | 68.4 | 88.4 | 65.2 | 28.2 | 80.3 | 48.0 | 71.2 | 75.7 | 57.7 | 89.3 | 80.0 |
> | 5% | 71.9 | 67.9 | 87.9 | 65.1 | 30.2 | 77.4 | 48.1 | 71.5 | 74.3 | 56.0 | 88.9 | 79.2 |
> | 10% | 71.6 | 67.4 | 87.4 | 64.4 | 30.3 | 77.0 | 47.4 | 72.0 | 73.2 | 55.4 | 88.4 | 78.1 |
> | 15% | 71.0 | 66.5 | 86.7 | 63.4 | 30.1 | 76.4 | 46.7 | 72.2 | 70.8 | 54.3 | 87.6 | 76.9 |

---

> ### Author Response · Authors · 2025-11-24
> **Response to Reviewer cXEa Q4**
>
> **4. How do the authors envision handling the omission of background information in domains where background context could be a valuable cue (e.g., occlusions, instance support)?**
>
> Referring to the response to the second weakness, we think the informative background voxels are those located on the border of the instance in the occlusions and instance support cases. In contrast, voxels far away from the foreground contribute little to the accuracy of bounding box localization. To address potential boundary ambiguity, we adopt a box enlargement strategy in the foreground scoring supervision. This enlargement effectively incorporates the near-object background regions that may contain valuable geometric cues, while excluding distant background points that are unlikely to provide meaningful information.

---

> ### Author Response · Authors · 2025-11-24
> **Response to Reviewer cXEa Q5**
>
> **5. Please elaborate on any observed failure modes (missed detection, false positives/negatives), especially with reference to dataset bias, rare classes, or environmental conditions.**
>
> Thank you for the suggestion regarding the inclusion of failure case analysis. We have visualized the results of our approach compared with previous works in the KITTI dataset, as illustrated in Figure 6 of the supplementary materials. It can be observed that the primary negatives and missing detections occur in the small or hard objects, which previous works also fail to recognize. We have also carefully visualized representative failure cases in the nuScenes dataset, as shown in Figure 7. The green boxes denote the ground-truth annotations, while the red and blue boxes represent the predictions of our model and previous methods, respectively. We observe that our approach occasionally produces false negatives on smaller object categories, such as pedestrians and barriers, and certain confusing scenarios may lead to false positives. Nonetheless, the previous method, LION, also frequently fails on these challenging cases. We have added the visualization of failure modes in the revision (highlighted in lines 920-925 of our revision).

---

> > ### Comment · Reviewer_cXEa · 2025-11-25
> >
> > Thanks for your efforts to addressing my concerns. I will maintain my score and vote for acceptance.

---

### Author Response · Authors · 2025-12-01
**Summary of Revisions and Responses to Reviewers**

We sincerely thank the reviewers for their suggestions and for recognizing the novelty and contributions of our work. In this summary, we provide comprehensive responses to the concerns raised regarding robustness, context preservation, and architectural design.

**Robustness Verification and Context Preservation.** A primary concern raised by Reviewers cXEa and 3Dbr involved the potential sensitivity to imperfect foreground scoring and background loss. We addressed this through rigorous empirical stress tests, simulating noise by substituting 5%–15% of sampled foreground voxels with background voxels. The model exhibits exceptional resilience, maintaining high performance with negligible degradation even at a 10% replacement ratio—a margin significantly exceeding typical real-world errors. Moreover, to resolve concerns about background omission (Reviewers cXEa and jtLi), we clarified our enlarged grounding box strategy (0.5m in X/Y-axes and 0.25m in Z axis). By expanding sampling boundaries and retaining global density information, our design explicitly preserves essential instance support and boundary cues while effectively filtering out irrelevant clutter.

**Architectural Justification and Efficiency Gains.** To substantiate our design choices (Reviewer cXEa), we benchmarked our framework against alternative causal sequence models, including RetNet, RWKV, and LSTM. Our Mamba-based design demonstrates superior efficacy, outperforming all alternatives (+0.7% mAP vs. RetNet, +1.3% mAP vs. RWKV), thus validating it as the optimal linear sequence model for this task. Addressing the efficiency-performance trade-off (Reviewer jtLi), we emphasize that our method breaks the conventional trade-off by achieving SOTA accuracy while simultaneously reducing encoder FLOPs by 43.7% and increasing inference FPS by 23.9%. This dual advantage represents a substantial advancement over prior global-encoding paradigms.

**Presentation Improvements and Failure Analysis.** Following Reviewer 7P9E's suggestions, we have revised the manuscript to decouple theoretical derivations from the inference pipeline clearly and elaborated on the motivation for the local token. Furthermore, to ensure transparent evaluation (Reviewers cXEa and 3Dbr), we provide detailed visualizations of failure cases and qualitative comparisons in the supplementary material, offering deeper insight into the model’s behavior in challenging scenarios—where all the compared methods also fail.

The reviewers have acknowledged the novelty and effectiveness of our foreground-focused paradigm and overall framework. With the added robustness analysis, expanded comparative experiments, and transparent discussion of limitations, we believe we have thoroughly addressed all concerns regarding the method’s reliability and performance.

---

### Meta-Review · Area_Chair_UCqA · 2026-01-04

**Summary:**

This work presents Fore-Mamba3D, a new Mamba-based method that focuses on effective linear encoding of foreground features to achieve better 3D detection performance. It introduces a regional-to-global sliding window strategy and the SASFMamba module. The model not only achieves performance improvements on multiple datasets, but also shows significant advantages in inference efficiency.

**Reviewer Concerns:**

The authors’ responses have addressed most of the reviewers’ concerns. For example, the authors clarified the rationale behind the architectural design raised by Reviewer cXEa, responded to Reviewer 3Dbr’s concern regarding the perceived lack of novelty of the proposed method, addressed Reviewer jtLi’s questions about the trade-off between performance improvement and inference efficiency, and explained the motivation for introducing the local token as questioned by Reviewer 7P9E.

**Reviewer Scores:**

Currently, there are three reviewers give the negative scores. Because most of concerns have been addressed. I believe that, most of reviewers will keep or raise their scores.

---

### Decision · Program_Chairs · 2026-01-26

Accept (Poster)